# OV Modulators of the Paediatric Brain TIME: Current Status, Combination Strategies, Limitations and Future Directions

**DOI:** 10.3390/ijms25095007

**Published:** 2024-05-03

**Authors:** Konstantinos Vazaios, Ronja E. van Berkum, Friso G. Calkoen, Jasper van der Lugt, Esther Hulleman

**Affiliations:** Princess Máxima Center for Pediatric Oncology, 3584 CS Utrecht, The Netherlands; k.vazaios@prinsesmaximacentrum.nl (K.V.); f.g.j.calkoen-2@prinsesmaximacentrum.nl (F.G.C.); j.vanderlugt@prinsesmaximacentrum.nl (J.v.d.L.)

**Keywords:** oncolytic viruses, paediatric brain tumours, immunotherapy, clinical trials, immune-oncology

## Abstract

Oncolytic viruses (OVs) are characterised by their preference for infecting and replicating in tumour cells either naturally or after genetic modification, resulting in oncolysis. Furthermore, OVs can elicit both local and systemic anticancer immune responses while specifically infecting and lysing tumour cells. These characteristics render them a promising therapeutic approach for paediatric brain tumours (PBTs). PBTs are frequently marked by a cold tumour immune microenvironment (TIME), which suppresses immunotherapies. Recent preclinical and clinical studies have demonstrated the capability of OVs to induce a proinflammatory immune response, thereby modifying the TIME. In-depth insights into the effect of OVs on different cell types in the TIME may therefore provide a compelling basis for using OVs in combination with other immunotherapy modalities. However, certain limitations persist in our understanding of oncolytic viruses’ ability to regulate the TIME to enhance anti-tumour activity. These limitations primarily stem from the translational limitations of model systems, the difficulties associated with tracking reliable markers of efficacy throughout the course of treatment and the role of pre-existing viral immunity. In this review, we describe the different alterations observed in the TIME in PBTs due to OV treatment, combination therapies of OVs with different immunotherapies and the hurdles limiting the development of effective OV therapies while suggesting future directions based on existing evidence.

## 1. Introduction

Among childhood cancer, tumours of the central nervous system (CNS) remain the leading cause of death [1]. According to World Health Organisation (WHO) Classification for Paediatric Brain Tumours (PBTs), the main tumour entities consist of high-grade gliomas (HGGs), low-grade gliomas (LGGs), medulloblastomas (MBs), ependymomas (EPNs), atypical teratoid/rhabdoid tumours (ATRTs) and embryonal tumours with multilayered rosettes (ETMRs), each demonstrating a very distinct molecular background and cell of origin and each being further divided into additional subtypes [2]. Overall survival rates (OSRs) of PBTs are highly variable and depend on the type of tumour, grade, location, size and age of the patient and the ranges can vary from <30% at 24 months for HGGs to >80% at 60 months for LGGs [3,4]. The current standard of care for these brain tumours comprises a combination of surgical resection (if operable), radiotherapy (RT) and chemotherapy [1]. Every PBT type is treated differently depending on the location and grade; however, in all cases, clinicians choose maximal resections when possible, higher doses of RT and chemotherapy, leading to increased survival of the patients [4]. However, the increased survival comes at the cost of debilitating long-term side effects such as endocrine dysfunction and a decline in neurocognitive development for patients at a younger age in particular [4,5]. The limited success and the high risk of long-term neurological effects are all indicative of the need for more effective and targeted forms of treatment.

Cancer immunotherapies have been rapidly developing in terms of efficacy and are becoming crucial in the treatment of tumours due to their specificity to tumour cells while leaving healthy cells intact. These immunotherapies entail all forms of treatment that use the immune system to target malignant cells and have been successfully used in the treatment of multiple hematologic malignancies and solid tumours [6]. Examples include antibodies, immune checkpoint inhibitors (ICIs), vaccines and chimeric antigen receptor (CAR)-T cells [6]. Considerable obstacles in the use of immunotherapies in PBTs have been the low tumour mutational burden (TMB), the blood–brain barrier (BBB) and the common presence of a ‘cold’ tumour immune microenvironment (TIME) [7,8,9].

A major obstacle to effective immunotherapy in the paediatric TIME is the presence of a low TMB with a mutation frequency of 14 times lower than in adult tumours and with 0–6 somatic coding point mutations per megabase of targeted territory [8,9]. This is a major disadvantage for designing effective immunotherapy as the amount of potentially druggable tumour-specific antigens (neoantigens) is expected to be much lower.

The BBB is a physical barrier that greatly limits vesicular transport while allowing only passive transport by small (<400 Da) or lipid-soluble molecules in an effort to protect the brain from toxins and pathogens [10]. This barrier in PBTs demonstrates a great heterogeneity in integrity among the different subtypes of tumours while remaining less permissive than adult counterparts, which can greatly reduce the influx of systemically administered therapeutic drugs, their bioavailability and clinical response [11].

A suppressive or ‘cold’ TIME consists of low expression of immunogenic markers on the tumour cells, low infiltration of proinflammatory lymphocytes, an abundance of regulatory immune cells, and the presence of tumour-associated macrophages/microglia (TAM) that hamper effective immunotherapeutic strategies [3,12]. Provided that an immunosuppressive microenvironment in cancer patients correlates with poor prognosis and limits the effectiveness of immunotherapies, strategies to switch from an immunosuppressive to proinflammatory microenvironment may be key for robust and long-term anti-tumour responses required for therapeutic success [13].

One potential strategy is virotherapy where the use of oncolytic viruses (OVs) is harnessed to specifically target and kill tumour cells [14]. A limited number of PBT clinical trials using OVs resulted in a number of cases with improved median overall survival (MOS), low-grade adverse events (AEs) and lack of neurological deterioration, thus sparking an interest in their potential therapeutic value [15,16,17,18]. In addition, OVs have the innate ability to induce a systemic and local immune response. This ability can be further enhanced through genetic modifications [14]. The OV-driven immune response is able to elicit an anti-tumour effect by stimulating the TIME [14]. The shift towards a pro-inflammatory TIME by the OVs creates windows of opportunity for effective treatment with other immunotherapeutic modalities. Therefore, coupling OVs with immunotherapies like ICIs, adoptive cell therapies or tumour antigen vaccines may be the future of effective therapeutics in PBTs. Early investigations of these combinations have demonstrated the synergistic effects of OVs with immunotherapies, leading to improved immunomodulation and survival benefits in adult clinical trials [19,20].

Even with the accumulation of positive clinical responses, investigations on the immunomodulatory effects of OVs remain unexplored either due to the limitations of existing pre-clinical models or ineffective monitoring. This review aims to provide an overview of the current knowledge on the modulation of the TIME by OVs in PBTs, modifications employed for an enhanced immunomodulatory effect and combination strategies with existing immunotherapies. Furthermore, we discuss the current barriers to the development and investigation of effective OV therapies and subsequent combinations while providing suggestions to overcome those. Lastly, we provide suggestions on different clinical aspects affecting OV clinical trials in need of optimisation.

## 2. Paediatric Brain Tumour Immune Microenvironment

All PBTs were considered highly immunologically ‘cold’ in the past, with new insights demonstrating that each entity of a PBT is characterised by a varying immunophenotype that results in heterogenous tumour microenvironments (TME) (Figure 1) [12]. More investigations associated the TIME heterogeneity as a predictor of prognosis and survival [21]. These differences were not only observed between entities of tumours but also extents among molecular subgroups within one tumour entity [22,23]. Briefly, LGGs and ATRTs bear higher levels of infiltration of myeloid cells and lymphocytes than MBs and HGGs [23,24], whereas in HGGs and diffuse midline gliomas (DMG) lymphocyte infiltration is very low compared to other PBTs [25]. In this review, the most commonly occurring characteristics spanning the multiple PBT types will be discussed.

The lymphocyte compartment of the PBTs is characterised by a low infiltration of B cells, CD4+ helper T cells and CD8+ cytotoxic T cells [25]. Both T cell subsets are crucial in assisting and executing an adaptive immune response against tumour cells. Concurrently, regulatory T cells (Tregs) that release and maintain immunosuppressive cytokines like interleukin 10 (IL-10) and transforming growth factor-β (TGF-β) remain more prevalent in the tumour than in the surrounding healthy tissue [12,25]. Furthermore, increased levels of TGF-β and increased expression of immune checkpoint molecules like the programmed cell death ligand 1 (PD-L1) and B 7 homolog 3 protein (B7H3) on the surface of the tumour cells, especially on HGGs, result in the inhibition of effector T cells and, subsequently, in further Treg infiltration [25].

In PBTs, natural killer (NK) cells and other innate lymphocytes (NKT and γδ) that recognise target cells independently of human leukocyte antigen (HLA), have hindered effector functions due to the extremely low expression of activating receptors necessary for a NK cell-mediated anti-tumour response. Such receptors are the natural killer group 2D (NKG2D) receptor and UL-16 binding proteins (ULBP) 1 through 6 on NK cells within the TIME of MBs and HGGs [26]. One potential mechanism behind this low surface expression is thought to be induced by excessive TGF-β present in the TIME [27].

The myeloid compartment of the paediatric brain TIME is composed of regulatory cell types, such as myeloid-derived suppressor cells (MDSCs) and M2 macrophages (anti-inflammatory and tumour supporting) and effector cell types, such as dendritic cells (DCs) and M1 macrophages (pro-inflammatory and tumour inhibiting) [28,29]. The balance in the immunosuppressive TIME of PBTs sways more towards higher levels of MDSCs and M2 macrophages than to DCs and M1 macrophages [12].

A unique myeloid cell population of the brain known as microglia, also described as the macrophages of the brain, are multifunctional cells responsible for debris removal, release of neurotrophic and growth factors as well as the resident innate immune cells of the brain [30]. Like macrophages, microglia demonstrate an M2 tumour-supporting and anti-inflammatory state and an M1 pro-inflammatory and anti-tumour state [31]. Their phenotype seems to be correlated by the tumour entity with EPNs, with ATRTs having microglia with more proinflammatory phenotype while MBs and HGGs have an anti-inflammatory phenotype [24].

DCs function as antigen-presenting cells (APCs) as they can capture antigens, originating from tumours or infected cells and effectively “cross-present” them to other immune cells, especially to CD8 T cells [32]. However, in PBTs the limited presence of tumour-associated antigens (TAAs) due to the low TMB and the increased release of CD47 often detected in MYC-amplified tumours that reduce phagocytosis lead to low levels of antigen presentation and, consequently, lower anti-tumour immunity [33].

## 3. Oncolytic Viruses as Anti-Cancer Agents

### 3.1. Characteristics of OVs

OVs employ naturally occurring or genetically engineered viruses that selectively infect, replicate in and kill tumour cells while healthy cells remain unaffected [14]. They originate from different viral families. The most basic oncolytic virus is composed of a core and a capsid. The core of an oncolytic virus comprises either DNA or RNA, which can be either double-stranded (ds) or single-stranded (ss), linear or circular, storing their genetic information whilst concurrently driving characteristics. DNA oncolytic viruses have more efficient replication abilities due to high-fidelity polymerase [34]. On the other hand, they are less immunogenic as the hosts are usually equipped with deficient DNA-sensing machinery and present a small risk of viral DNA integrating into the host genome, potentially exacerbating tumour mutations (e.g., activating oncogenes) [34]. RNA oncolytic viruses have enhanced delivery efficiency and a distinct advantage in targeting CNS tumours [34]. This is usually attributed to their size being considerably smaller than DNA viruses, enabling more efficient crossing of the BBB [35]. The most extensively studied viruses that have been evaluated in clinical trials of brain cancers are described in Table 1 and Figure 2A [36,37,38].

Wild-type and genetically modified OVs recognise overexpressed or tumour-specific surface receptors and abnormal upregulated or downregulated signalling pathways and their related cellular products unique to tumour cells [36]. Moreover, the high rate of metabolism prevalent in tumour cells provides an attractive target for efficient viral replication [37]. Some common modifications for improved viral entry include (a) replacement of the original viral fibre with a viral fibre of the desired tumour tropism (serotype switching), (b) modification of viral capsid proteins or viral envelope glycoproteins for improved viral entry and (c) insertion of genes encoding a single-chain antibody specific to a known tumour-surface antigen [36]. Different strategies to improve tumour specificity based on intracellular tumour features are proposed. Insertion of tumour-specific promotors in the viral genome increased specificity by allowing selective expression of viral genes in tumour cells. Mutation or knock-out of specific viral genes deprives OVs of their ability to replicate/virulence in normal cells. This can be used to achieve selective OV replication in tumour cells that are characterised by loss of, e.g., tumour-suppressive genes (Figure 2B) [36,37].

### 3.2. Mechanism of Action of OVs

After successful entry into the tumour cells, the anti-tumour effect of OVs can be divided into two main mechanisms. First, OVs can replicate and lyse tumour cells to release new viral particles into the TME, leading to successive cycles of infection [37,49]. Second, tumour cell lysis results in the release of TAAs, viral pathogen-associated molecular patterns (PAMPs), damage-associated molecular patterns (DAMPs) as well as chemokines and toll-like receptor (TLR) agonists [49].

PAMPs consist of common motifs expressed by infectious agents (e.g., viral capsids or nucleic acids), whereas DAMPs are usually derived from the host cell [49]. Examples of DAMPs include heat shock proteins, high mobility group box 1 (HMGB1) protein, calreticulin and ATP [49]. Members of both groups can bind to pattern recognition receptors such as TLRs on macrophages and NK cells and instigate type I interferon (IFN) signalling along with further innate immune responses [49]. Furthermore, the release of PAMPs, DAMPs, cytokines and TAAs into the TIME results in the increased recruitment of NK cells and APCs such as macrophages and DCs into the tumour environment [50]. The secretion of pro-inflammatory mediators leads to the maturation and activation of APCs [50]. The presentation of TAAs by APCs can activate adaptive immune cells either locally or in the lymph nodes, resulting in lymphocyte infiltration of activated CD4+ and CD8+ T cells into the tumour [50]. In the tumour, the cytotoxic CD8+ T cells can specifically eliminate tumour cells that express the TAA via major histocompatibility complex (MHC) class I, regardless of whether they were infected with the OV [49,50]. Moreover, memory T cells are formed, improving anti-tumour immunity against future tumour challenges. Thereby, OVs can kickstart a broad anti-tumour immune response by providing immunogenic TAAs and inducing increased levels of infiltrating immune cells (Figure 2C) [50].

## 4. Paediatric Brain Tumour Immune Microenvironment Modulation by OVs

### 4.1. Preclinical Evidence

Preclinical evidence in different PBT models demonstrates the ability of OVs to modulate the brain TIME. Here, we provide a summary of preclinical findings from PBT models (Figure 3, Table 2).

Immunocompetent MB mouse models treated intratumorally with C134, a modified HSV-1, demonstrated improved survival coupled with an increased influx of M1 macrophages, DCs, CD4+ and CD8+ T cells and NK T cells through single-cell analysis. Additionally, a significant increase in programmed cell death protein 1 (PD-1) and lymphocyte-activation gene 3 (LAG-3) cell expression was observed on CD4+ and CD8+ T cells as well as in the percentage of LAG3+ cells of both respective T cells, while PD-L1 was increased on the surrounding cells of the TME [51].

Further investigation of the gene expression of these infiltrating cells revealed altered expression of antigen presentation, IFN signalling, cytokine signalling and cytotoxicity-related genes. More specifically, in lymphocytes (T, B and NK) expression of genes coding for granzyme A, B and natural killer cell granule protein 7 (NKG7) was increased. In monocytes, macrophages and microglia, an increased expression of cytokines, chemokines and MHC I genes was observed, while genes related to IFN-response and chemokines in DCs were increased [51].

A prior study investigating malignant glioma treatment with C134 also demonstrated a significant increase in the number of CD8+ T cells, but not CD4+ T cells in the CSF of both models. Interestingly, in T cell-depleted mice C134 treatment did not increase survival. In addition, re-challenge for 70 days led to inhibition of tumour growth, indicating that a circulating memory immune response had developed in the mice after treatment with C134 [52].

A similar increase in CD4+ and CD8+ T cells post-treatment with an oncolytic AdV named Delta-24-RGD (DNX-2401) within glioma tumours of immune-competent Syrian hamsters was observed [53]. The effect of increased CD8+ T cells by Delta-24RGD was also demonstrated in immunocompetent humanised mouse models bearing ATRT tumours, though in this model a reduced presence of macrophages was observed [54]. Interestingly, in the same study, the ETMR-bearing mice demonstrated a pronounced activation and recruitment of ionised calcium-binding adapter molecule 1 (Iba-1)-positive cells (a marker for proinflammatory macrophages and microglia) at the tumour margins; however, no active viral infection or T cell infiltration was detected, indicating fast clearance of Delta-24RGD [54].

Regarding the pre-clinical evidence of interactions between OVs and macrophages and microglia, there is an extreme lack of research investigating this in paediatric tumours such as PBTs [55]. However, based on what has been studied in adult brain tumours such as glioblastoma (GBM), the role of macrophages and microglia on OV anti-tumour efficacy is multifaceted. Due to their traditional activation towards one of two polarisation states, the M1 pro-inflammatory and the M2 immunosuppressive state, M1 and M2 microglia and macrophages are generally considered to be anti-and pro-tumorigenic, respectively. In line with this, there is evidence that OV therapy can induce microglia and macrophage switching from an M2 to M1 phenotype in the GBM microenvironment [56]. As a result, pro-inflammatory M1 microglia or macrophages can produce soluble factors such as reactive oxygen species (ROS), nitric oxide (NO), tumour necrosis factor-alpha (TNF-α) and IL-1β. These factors have the potential to induce apoptosis, DNA damage or cytotoxicity, resulting in the direct elimination of tumour cells [57]. Indirectly, M1 TAMs can also recruit and activate other immune cells such as DCs, NK and T cells [57].

However, simultaneously, M1 pro-inflammatory microglia and macrophages form the primary anti-viral immune response through the production of type I IFNs and phagocytosis of OVs [58]. This leads to the accelerated clearance of OVs and thereby a decreased viral infection, replication and lysis of tumour cells. Thus, while M1-like macrophages are expected to boost virus-induced activation of the anti-tumour immune response, they may also facilitate early virus clearance. In contrast, M2-like macrophages, linked to tumour angiogenesis, metastasis and suppression of the anti-tumour immune response may also suppress the anti-viral immune response and support oncolysis [55].

Based on these findings, we cannot determine an exact mechanism for OV-mediated immune modulation that results in a potent anti-tumour response. Nevertheless, it appears that OVs can elicit durable innate and adaptive immune responses across various in vivo models of PBTs, often resulting in extended overall survival, which is lost in the absence of a functional TIME, indicating therapeutic efficacy and high dependency on the immune system.

**Table 2 ijms-25-05007-t002:** Overview of the OV-induced immune cell modulations reported in preclinical paediatric brain tumour models.

OV Type (Name)	OV Modifications	Tumour Type (Cell Line)	Animal Model	Survival	Immune Cell Type	Modulation	Year and Ref.
HSV (C134)	Deletion of both copies of the principal virulence gene γ_1_34.5. Additionally has IRS1 gene under control by human cytomegalovirus immediate early promoter.	Mouse-MB (CMYC, MYCN)	Syngeneic C57BL/6 mice	Improved	CD4+ T cells	Increased influx in TME, Increased expression of PD-1 and LAG-3	2023 [51]
					CD8+ T cells	Increased influx in TME, Increased surface expression of PD-1 and LAG-3 and increased expression of granzyme A & B genes	
					NK cells	Increased level in TME and increased expression of granzyme A, B and NKG7 genes	
					DCs	Increased influx in TME and increased expression of IFN and chemokine genes	
					Macrophages	Increased levels of M1 macrophages and increased expression of cytokine, chemokine and MHC I genes	
					Microglia	Increased expression of cytokine, chemokine and MHC I genes	
		Glioma (Neuroglial-Neuro2A, DBT mice glioma)	Syngeneic A/J mice and syngeneic Balb/C mice	Improved	CD8+ T cells	Increased influx in TME	2018 [52]
					Memory formation	Developed and capable of controlling rechallenge	
AdV (Delta-24-RGD)	Insertion of RGD-4C peptide in the fibre knob. 24 bp deletion in E1A viral gene responsible for Rb-binding.	Glioma (Ham GSCs)	Syrian Hamster	Improved	CD4+ T cells	Increased influx in TME	2021 [53]
					CD8+ T cells	Increased influx in TME	
		DMG (TP54, TP80, NP53), HGG (CHLA-03-AA, PBT-24)	Balb/C mice	Improved	CD4+ T cells	Increased influx in TME	2019 [59]
					CD8+ T cells	Increased influx in TME	
		ATRT (CHLA-06), ETMR (PFSK-1)	CD34-humanized NSG-SGM3 mice	Improved	CD8+ T cells	Increased ratio within CD3+ population	2021 [54]
					Macrophages	Pronounced recruitment and activation at tumour margins	
					Microglia	Pronounced recruitment and activation at tumour margins	

### 4.2. Clinical Evidence

There have only been a limited number of clinical trials involving oncolytic viruses in PBTs, some of which are still ongoing and pending results. However, here we review what has clinically been established in paediatric human trials so far, with a focus on OVs reported to be modulating the TIME (Table 3, Figure 2) [60].

In a dose-escalation study of DNX-2401 in patients with newly diagnosed DMG, it was found in tumour tissue from patients that immune cell infiltration of CD4+ and CD8+ T cells was scarce while CD11b+ myeloid cells were the most abundant immune population upon diagnosis [15]. Tumour tissue from one patient demonstrated increased numbers of CD4+ and CD8+ T cells along with an absence of Treg cells and a reduced number of CD11b+ cells after administration of DNX-2401. While upregulation of gene ontology (GO) terms related to viral processes and immune response were demonstrated in TAMs. However, upon later autopsy of this patient, the infiltration of effector lymphocytes had decreased and was replaced by an increased number of immunosuppressive macrophages [15]. The patients from this study demonstrated an MOS of 17.8 months and OSR of 50% at 18 months and their progress-free survival was linked to the T-cell receptor (TCR) clonality in their peripheral blood mononuclear cells [15].

A similar strong increase in infiltrating lymphocytes was seen in a Phase 1 study in children and adolescents with recurrent or progressive HGG after treatment with the genetically engineered HSV G207, leading to a median OSR of 12.2 months [16]. More specifically, matched resection tissue from pre-treatment and 2–9 months post-treatment showed a significant increase in CD4+ and CD8+ T cells within the tumour as well as areas adjacent and distant from where G207 was inoculated [16]. Interestingly, the tumour-infiltrating lymphocytes were observed at later timepoints, coinciding with the absence or lack of G207 replication, indicating a persistent immune response after OV clearance [16]. Furthermore, the presence of B lymphocytes and plasma cells pre-treatment was low, whereas post-treatment, levels of B lymphocytes were increased in one patient and levels of plasma cells in two patients [16]. Interestingly, when comparing pre-treatment and 3 months post-treatment tissue biopsies from a patient from the same trial, significantly increased expression of cytotoxic T lymphocyte-associated protein 4 (CTLA-4) and PD-1 checkpoint molecules was observed on the infiltrating CD8+ T cells [61]. Moreover, local tumour cells significantly expressed increased amounts of PD-L1 and IDO after treatment with G207 [61]. These data suggest that along with OV-induced immune cell infiltration of the TME, there is an increase in immune checkpoint molecules on tumour and immune cells [61].

A significant increase in tumour infiltrating CD8+ T cells was also observed in one patient when comparing pre- and post-treatment tissue, in a phase 1b trial studying the safety of recombinant PV lerapolturev (formerly known as PVSRIPO) in recurrent paediatric HGG [18]. However, the patient was pre-treated with an inactivated PV vaccine booster 1 week prior to lerapolturev [18]. As a result, it was unclear whether this CD8+ T cell infiltration was the effect of the pre-treatment booster or lerapolturev itself [18].

Pelareorep is an oncolytic RV which had shown preclinical efficacy in immunocompetent glioma models after intravenous injection when combined with preconditioning with granulocyte-macrophage colony-stimulating factor (GM-CSF), which recruits APCs to the TIME [62]. In a phase 1 trial of Pelareorep in paediatric patients with recurrent or refractory high-grade brain tumours led to a MOS of 11.7 months with increased levels of monocytes observed in 60% of patients post-treatment [17]. Moreover, all patients showed drops in white blood cells, platelets and neutrophils in the first week after therapy, which recovered in week two [17]. Serum cytokine levels showed no significant differences between patients and it is unclear whether these findings were the result of preconditioning with GM-CSF or the treatment with pelareorep [17]. The findings of this study, however, were rather limited by the level of pretreatment and prognostic uncertainty of the included patient cohort, as well as the fact that only serum samples were taken for assessment of immune response [17].

In the paediatric clinical trials described above, OV treatment was related mainly to Grades 1, 2 and in some rare cases Grade 3 AEs and with 1 Grade 4 case (pelareorep) and thus did not lead to any serious neurological deterioration and decline of quality of life (QoL) by the time the clinical trials concluded [15,16,17,18]. However, most OV clinical trials do not have data available on QoL, neurological changes or long-term effects from long-term survivors due to how recently those trials were initiated, and future clinical trials on OVs should accommodate for this.

In light of these observations, these clinical trials suggest that oncolytic viruses may modulate the TIME by increasing immune cell infiltration, particularly of CD4+ and CD8+ T cells. Especially in these cases where immune modulation was successful, the survival benefit was significant. However, further research is needed to better understand the specific mechanisms involved and to determine whether these effects are primarily due to the viruses themselves or other factors such as pre-treatment interventions.

### 4.3. Armed OVs

As more and more evidence demonstrates the immunomodulatory actions of OVs and their importance for improved survival, genetic modifications began focusing on arming OVs with transgenes to enhance their effector functions [50]. These types of enhanced OVs have not been extensively studied in PBTs yet; therefore, the following data will be based mainly on adult brain tumour models or patients in order to demonstrate their immunomodulatory potential.

#### 4.3.1. Immunomodulatory Cytokines

In order to expand the innate capacity of OVs to stimulate an anti-tumour immune response, arming them with immunostimulatory cytokines to be expressed and released during infection and/or lysis of the tumour cells increases their concentration in the TME in order to help recruit immune cells and boost their anti-tumour efficacy [36].

For example, M032 a modified HSV expressing IL-12 is currently under investigation in an ongoing clinical trial in adult patients with recurrent or progressive malignant glioma (NCT02062827) [20]. The secretion of IL-12 by the infected tumour cells, led to IFN-γ production, enhancing the anti-tumour effect of cytotoxic T cells and NK cells [20]. R-115, an HSV, was also modified to express IL-12, inducing the same effects in the glioma mouse model [63]. Another engineered OV, Ad-TD-nsIL12, is an Adv engineered with a non-secretory form of IL-12 to localise its presence in the TME, thereby limiting adverse effects [64]. Ad-TD-nsIL12 is currently under investigation in primary and progressive paediatric DMG patients in a Phase 1 trial (NCT05717712 and NCT05717699).

OV-IL15C is an HSV that induces the expression and secretion of a soluble human IL-15/IL-15Rα and was successful in improving the survival in vitro and in vivo in models of adult GBM while enhancing NK-cell and CD8+ T cell cytotoxicity [65].

#### 4.3.2. Co-Stimulatory Molecules

Similar to the function of cytokines effective T cell activation requires interaction and stimulation of co-stimulatory molecules. Therefore, the modification of OVs to express immune co-stimulators could enhance the antigen presentation of tumours and lead to increased activation of tumour-specific T cells. Delta-24-RGDOX also known as DNX-2440 is an example of such an OV [66]. It is an oncolytic Adv that expresses OX40 ligand (OX40L), on the surface of a panel of human and mouse tumour cell lines [66]. Moreover, Delta-24-RGDOX led to higher levels of CD4+ and CD8+ T cell infiltration and activity in the tumour environment than Delta-24-RGD. This improvement was negated by OX40L blocking [66]. Importantly, treatment with Delta-24-RGDOX demonstrated anti-glioma activity and a longer median survival rate than Delta-24-RGD-treated mice, demonstrating the advantage of boosting the immunomodulating effect of Ovs [66]. DNX-2440 is currently under a Phase I clinical trial (NCT03714334) in patients with recurrent GBM.

#### 4.3.3. Tumour Suppressor Genes

Similarly, common tumour suppressor genes can be engineered into Ovs to be synthesised and secreted by the infected tumour cells. An example of this is HSV-P10, which expresses the gene for phosphatase and tensin homolog deleted on chromosome 10 alpha (PTENα) [67]. IC treatment of mice with this OV led to increased immune cell recruitment of microglia, DCs, NK cells and CD8+ T cells into the tumour [67]. Treatment with HSV-P10 in comparison to the non-PTEN expressing HSVQ in a panel of tumour cell lines showed a reduction in cell surface PD-L1 expression on tumour cells, thereby helping to overcome tumour immune escape [67]. The therapeutic efficacy of HSV-P10 has additionally been validated preclinically in a model of GBM stem-like cells [68]. However, in this study, the interactions with the TME were not investigated yet.

## 5. Combinational Treatment of OVs with Immunotherapies

Over the years, immunotherapies have reached the status of being an important modality in the fight against different types of cancer. However, due to challenges such as the suppressive TME present in many solid tumours, including PBTs, the efficacy of immune monotherapy has not been optimal. The ability of oncolytic virotherapy to modulate the TIME towards a more active, immune ‘hot’ state provides potential for improved efficacy of other immunotherapies if used in combination with OVs (Figure 4).

### 5.1. Immune Checkpoint Inhibitors

The objective of immune checkpoint blockade (ICB) therapy is to disrupt immunosuppressive tumour signals and reinstate robust anti-tumour immune responses by the tumour-specific T cells. ICIs are antibodies that bind to checkpoint receptors and ligands, limiting their interaction, and, consequently, the downstream immune response inhibition. Examples of such receptors and ligands include PD-1 and its ligand PD-L1, CTLA-4 and its ligands CD80/86, LAG-3 and its ligands galectin-9 and MHC-II, and T-cell immunoreceptor with Ig and ITIM domains (TIGIT) and its ligands CD155/112. ICIs have been studied as monotherapy in many tumours before; however, low levels of tumour infiltrating lymphocytes (TILs), low mutational burden and low expression of TAAs hinder their efficacy in PBTs [69].

As mentioned before, treatment with the OV G207 induced the expression of CTLA-4 and PD-1 on infiltrating CD8+ T cells as well as PD-L1 and IDO on tumour cells [61]. This represents the basis of a potentially synergistic combination of OV therapy and ICIs. As of yet, there have not been any paediatric clinical trials using combinational therapy of both ICIs and OVs in brain tumours. However, there is much preclinical evidence supporting this combination as well as (pre-)clinical evidence from adult brain tumour patients.

Recently, a multicentre Phase 1/2 study evaluated the combination of intra-tumoral delivery of DNX-2401 and the intravenous anti-PD-1 antibody pembrolizumab in recurrent adult GBM. The overall survival at 12 months in this trial was 52.7%, whereas the overall survival at 12 months in a previous trial of DNX-2401 monotherapy was 32% [19]. Of note, no monotherapy comparator cohort for DNX-2401 or Pembrolizumab alone was included in this study [19]. The combination of both treatments was very well tolerated in all patients with AEs of Grade 3 and lower, demonstrating the safety of such combinations [19].

The therapeutic modalities of ICIs and OVs can also be combined into one therapeutic modality. OVs that encode for and secrete immune checkpoint inhibitor antibodies within the TME have been engineered. Moreover, OVs can be engineered to secrete soluble forms of immune checkpoint ligands, to act as decoy ligands and sterically hinder the binding of the receptor to its primary ligand on the effector immune cells [70,71]. Engeland et al. constructed an attenuated MV vector expressing the genes for antibodies against both CTLA-4 and PD-L1 [72].

Another alternative could be the targeting of “do not eat me” molecules expressed by the tumour cells (e.g., CD47 and CD24) to increase phagocytosis and antigen presentation for improved adaptive immunity [73]. An oncolytic AdV armed with SIRP-a-Fc or Siglec-10-Fc was able to express the extracellular domain of those ligands that can bind to the receptors CD47 and CD24, respectively [73]. Glioma mouse models rich with macrophages in the TME were treated with these oncolytic AdV, which led to improved anti-tumour activity [73]. This activity was attributed to the increased levels of phagocytosis and antigen presentation while reducing the proportion of MDSCs, TAMs and Tregs [73].

### 5.2. Adoptive Cellular Therapy

#### 5.2.1. CAR-T

CAR-T cell therapy is a form of adoptive cellular therapy where the patient (or donor)-derived T cells are modified to express a chimeric receptor, the CAR, capable of recognising specific surface tumour antigens and amplified ex vivo [74]. Upon reinfusion, these engineered CAR-T cells enter the TME, recognise and kill tumour cells that express the specific antigen [74]. However, CAR-T cell monotherapy encounters multiple barriers that can restrict its efficacy causing limited expansion and persistence in vivo [75]. Briefly, in solid tumours such as PBTs, the CAR-Ts have to overcome restricted tumour infiltration and low levels of chemokines. The presence of immunosuppressive cytokines and inhibitory ligands, as well as low expression of TAAs of the “cold” TME usually result in an impaired effector CAR-T cell phenotype [76]. As OV-mediated infection and lysis of tumour cells can lead to the release of type I IFNs, DAMPs, cytokines and chemokines, this may improve the aforementioned issues related to ineffective CAR-T cell therapy [77].

In addition to the favourable virus-intrinsic effects of OVs on the TIME for CAR-T cell therapy, OV engineering can be used to extend this favourable profile. As previously discussed, armed OVs can carry transgenes for chemokines and cytokines that promote CAR-T cell infiltration, proliferation and activity [78]. Moreover, in situ transgene production of ICI antibodies carried by OVs could limit local CAR-T cell inhibition in the TME [79]. An additional strategy under investigation involves using OVs for the targeted delivery of surface antigens to tumours. In this approach, the OV delivers the transgene coding for the antigen recognised by the CAR exclusively to tumour cells, leading to de novo cell surface expression of the antigen. This addresses the challenge of limited TAA expression and tumour recognition by CAR-T cells [80].

The use of Ad5delta-24 in combination with GD-2 CAR-T cells in PDX neuroblastoma mouse models improved survival while enhancing the persistence of the CAR-Ts in vivo compared to the GD-2 monotherapy [78]. When the OV was further modified to release RANTES and IL-15 from the infected cells and combined with GD-2 CAR-Ts, the survival benefit and CAR-T persistence were significantly higher compared to the combination of GD-2 CAR-Ts and Ad5delta-24 [78].

However, OV-activated immune responses do not necessarily all positively impact CAR-T cell therapy. In a study using a B16EGFRvIII tumour mouse model, the combination of a vesicular stomatitis virus (VSV) with IFNβ transgene VSV-mIFNβ, along with an EGFRvIII CAR-T cell therapy, was studied [81]. It was observed that VSV infection generated chemokine expression of CXCL10 and CCL5, which should be favourable for CAR-T cell recruitment [81]. However, against expectations, no therapeutic effect was observed, along with a large reduction in CAR-T cell numbers. A similar but more moderate reduction in CAR-T cells was observed after combining CAR-T cell therapy with RV [81]. It was discovered that the exposure of CAR-T cells to high levels of type I IFN (such as IFNβ) released upon OV infection led to the upregulation of PD-1, T-cell immunoglobulin and mucin domain-containing protein 3 (TIM-3), LAG-3 and FAS, leading to apoptosis due to exhaustion. Contrastingly, the combination of VSVmIFNβ with type I IFN-insensitive CAR-T cells (IFNAR1-knockout) led to reduced tumour growth and increased survival in a lymphodepleted mouse model [81]. Thus, high levels of type I IFN may be both beneficial for recruitment and activation of T cell responses as well as detrimental to T cells through activation of negative feedback mechanisms to prevent autoimmunity. Therefore, the combination of OVs with CAR-T cell therapy should be well balanced in order to ensure the resulting TIME is neither suppressive to T cells nor incendiary, leading to T cell damage [77].

For effective combined OV and CAR-T cell therapy in PBTs, suitable CAR targets should be evaluated. Multiple CAR-T cell therapies are currently under clinical investigation for PBTs. GD-2 targeted CAR-T cells are studied in DMG (H3K27M mutant) [82], HER2 CAR T cells in recurrent and progressive EPN [83] and B7-H3 CAR T cells in paediatric HGG [84]. Furthermore, preclinical studies are looking into targeting GPC-2, which is highly expressed in MB and HGG [85].

#### 5.2.2. NK and CAR-NK

NK cells act as a first line of response against viruses as part of the innate immune response and are equipped with receptors, namely NKp46, NKp30, NKp44, NKG2D and DNAX accessory molecule 1 (DNAM-1), which recognise ligands usually expressed on virus-infected cells [86]. Clinical data have demonstrated the activation of NK cells induced by the OVs, and combinations of OVs and adoptive NK cell therapy are currently being investigated [87]. In the case of NK and CAR-NK cells, the anti-viral response and thus preference for virally infected cells might lead to fast viral clearance before they are able to provide significant oncolysis and immune modulation. In light of this, many strategies are investigated. An example would be to deplete NK cells prior to OV treatment followed by adoptive NK treatment, a therapeutic sequence that significantly improved the survival of GBM-bearing mice [88].

Similar to CAR-T cells, CAR constructs can be expressed by NK cells to enhance their specificity to certain antigens. CAR-NK cells are not HLA alloreactive and thus can be derived from any donor without prior HLA matching enabling an off-the-self allogeneic therapy approach [89]. A study by Ma et al. compared the efficacy of the combination of a modified HSV, OV-IL15C and EGFR-CAR-NK cells in immunocompetent GBM-bearing mice [65]. The rationale was that alongside the virus-intrinsic effects of the OV, IL-15 would contribute to the development, homeostasis, activation and survival of the CAR-NK cells [65]. The combined strategy suppressed tumour growth and led to significantly longer survival than either therapy alone [65]. Additionally, the combination induced increased levels of infiltrating NK and CD8+ T cells and longer persistence of the CAR-NK cells [65]. Many CAR-NK cell therapies are preclinically investigated for GBM (HER-2, GD-2, c-Met, EGFR, EGFRvIII, B7-H3, AXL, CD73 and NKG2DL) with specifically GD-2-CAR-NK cells being evaluated against multiple paediatric DMGs [90]. The first clinical trial with CAR-NKs against GBM employs the HER-2 CAR (NCT03383978).

#### 5.2.3. CAR-Macrophages

In the case of macrophages, a CAR redirects the phagocytotic action of the macrophages towards specific antigens, and the CAR-macrophages (CAR-MΦs) continue to act as APCs, resulting in stimulation of the adaptive immune response. Recently, a study by Chen et al. showed that co-delivery of anti-CD47 and CD133-CAR-MΦs in GBM mouse models could significantly improve tumour regression and survival compared to the untreated mice [91]. The combination treatment increased the percentage of DCs, CD8+ and CD4+ T cells in the TME, increased expression of TNFα and IFN-γ, followed by an enrichment of M1 macrophages [91]. No published study has investigated the combination of CAR-MΦs with OVs; however, modified OVs could be used to express the targets for the CAR-MΦs or ligands that could block the tumour escape from phagocytosis such as CD47 and CD24 [71]. This would allow unhindered phagocytosis of the tumour and consequent antigen presentation from the CAR-MΦs and activation of adaptive immunity.

### 5.3. Cancer Vaccines

Cancer vaccines take various forms, such as DCs, peptides, nucleic acids or viral vectors used to stimulate or amplify an immune response against a specific tumour antigen by inducing immune memory against subsequent tumour growth. The low immunogenicity of PBTs is considered one big hurdle required to be overcome for effective action of cancer vaccines. This mainly stems from the limited epitopes presented on MHC II complexes on APCs in the TIME and the subsequent failure to recruit enough T (helper) cells in order to enact the cytotoxic action of anti-tumour T cells [92]. As such, the use of OVs alongside cancer vaccines is being developed in multiple directions. First, as immunoadjuvants, OVs can be used as a primary immune boost, which preconditions the tumour for the effective application of a tumour vaccine [93]. Second, OVs can become ‘oncolytic vaccines’ when one or more TAAs are encoded into the OV that will be presented later by the APCs to the T cells [93]. However, the concurrent endogenous expression of TAAs in healthy tissues poses an important concern for the safety of this type of treatment. Therefore, to optimise this treatment modality, tumour-specific antigens (neoantigens) should be identified.

### 5.4. Antibody Therapeutics

Bispecific T cell engagers (BiTEs) consist of two antibodies connected with a flexible linker. One of the two usually targets a TAA on the tumour surface, whereas the other targets CD3 or a different cell-surface molecule on T cells (preferably an activator) [94]. Consequently, BiTEs bring tumour cells and T cells together, leading to polyclonal T cell activation, independent of MHC/TCR presentation and co-stimulation [94]. The poor delivery and penetration into solid tumours demonstrated by the BiTEs limit their therapeutic success [94].

One solution was to combine CARs with BiTEs into a single modified T cell product, the CAR.BiTE, which improves the specificity of the CAR-T towards the tumour. In vitro preclinical testing demonstrated favourable T-cell differentiation and phenotype compared to those activated by the CAR or BiTE alone [95]. The CAR-EGFRVIII.BiTE-EGFR were able to induce cytotoxicity in multiple PDX GBM cell lines expressing EGFRVIII and EGFR as well as EGFRVIII negative cells expressing only EGFR [96].

The other solution would be to use OV genetic carriers to secrete BiTEs upon arrival and infection of the tumour, bypassing delivery issues and promoting T cell activation in the TIME [97]. Little research has been conducted into the combined use of OVs and BiTEs in both paediatric and adult brain tumours. However, the continuous search for paediatric brain antigen targets suggests such a strategy would be possible in the future.

## 6. Overcoming Limitations and Obstacles of OV Therapy

### 6.1. Preclinical Barriers

Brain tumours in children exhibit distinct molecular and immunological characteristics compared to those found in adults. Thus, the extensive knowledge about adult brain cancers cannot be translated completely to PBTs [12]. Therefore, the applicability of effective immune-based therapies originating from findings in adult disease models should be carefully assessed in representative models of paediatric disease [25]. This task is more challenging than it may initially appear.

Brain tumours are commonly studied in human or patient-derived xenograft (PDX) mouse models due to their translational potential. Orthotopic PDX brain tumour models develop within an intracranial environment that closely resembles the physical and chemical matrix found in human brains. However, these tumours grow in immunodeficient mice lacking a host immune system. Thus, it is impossible to study an endogenous T cell response in such models. Even though PDX models have facilitated the study of various immunotherapies, the absence of key components within the TIME raises concerns regarding the translational validity of these studies. Alternative models, such as nude athymic mice, are incapable of mounting an adaptive immune response but retain functional subsets including B cells, DCs, macrophages, NK cells and innate immune responses [28]. Moreover, some immunocompetent models have been developed by introducing driver mutations that were identified in human tumours [28,98]. For example, an immunocompetent mouse model of DMG was successfully generated by Du Chatinier et al. [99]. They established tumour cell lines from DMG mouse models that had been induced through intra-uterine electroporation of DMG-specific mutations into the embryonic brainstem [99]. Hereafter, these primary murine tumour cells were orthotopically implanted as allografts in syngeneic mice [99]. This led to the generation of secondary brain tumours that accurately reflected the morphology and growth pattern of human DMG [99]. Importantly, the TIME of these model tumours mirrored the immune ‘cold’ TME that is observed in DMG patient material [99]. Humanised mouse models, on the other hand, are engrafted with brain tumours that originate from patient samples in a similar manner to PDX models, yet also engrafted with human immune cells [100]. Though this model can recreate most TIME interactions, they are unable to fully develop mature innate cells, they lack functional lymph node structures and germinal centres and their antigen-specific antibody response is greatly curbed [100].

Another issue related to the preclinical modelling of PBTs and therapeutic interventions is the differences observed between animal and human-derived models. For instance, in a study of oncolytic HSV in MB by Hedberg et al., major differences between animal-derived cell lines and human cell lines of MB were observed in terms of viral toxicity and viral recovery [51]. Human-derived cell lines were more sensitive to OV infection and OV-mediated killing than the mouse cells. Moreover, substantial increases in viral yield in the human cells were observed [51]. This highlights the need for accurate translational models, especially between species, to avoid cases of early discontinuation of potential therapeutics, due to underperformance demonstrated in murine cells as opposed to their intended human target.

In an attempt to avoid this underperformance and reduce the use of animal models, multiple ex vivo models are being created using patient-derived cells. These models vary in complexity from simple mono-culture spheres to complex organoids, with their common trait being the maintenance of the phenotype and characteristics of the tumour of origin [101,102]. The use of those in vitro techniques has enabled the investigation of the oncolytic efficacy of OVs against multiple types of PBTs and enable the investigation of OV interactions with a limited number of immune cell populations [103,104]. However, with the current technology and knowledge, those models remain insufficient at fully recreating all the events that occur during the treatment and cannot accommodate the multiple factors that would, in practice, affect the efficacy of the treatment.

As a solution, some scientists suggest that a variety of models should always be used complementarily as factors that are absent or non-reactive in one model could be assessed in another to obtain a more complete image of the ongoing interactions e.g., the use of multiple animal species or combinations of in vivo and in vitro models [77]. In an ideal scenario, the advancement of preclinical models would encompass a wide spectrum of PBTs, including their subtypes, various disease stages, and diverse local TIME.

### 6.2. Monitoring Barriers

In the field of OV therapy, accurate monitoring of viral delivery and characterising the anti-tumour immune responses triggered by the virus in a reliable, non-invasive manner is difficult.

Currently, clinical efficacy is determined using a combination of magnetic resonance imaging (MRI) and blood tests. However, in MRI imaging the OV-induced influx of immune cell infiltration can increase the tumour volume, leading to a perceived pseudoprogression on the imaging, a problem reported by clinicians in all main paediatric clinical trials [15,16,17,18]. This compromises our ability to accurately track the extent of progression-free survival as well as the progression of disease based on imaging [15]. One proposed solution to this issue is the implementation of the durable response rate (DRR) as an endpoint in cancer immunotherapy, especially in clinical studies. As a response may occur after an initial increase in tumour size, the need for a durability dimension in the measurement of response became apparent. The DRR is defined as the rate of objective response (complete response/partial response) by WHO criteria, lasting ≥6 months continuously beginning within the first 12 months after initiation of treatment. Importantly, disease progression is allowed prior to the onset of response [105]. Attaining a lasting response correlated with clinical advantages, including improved overall MOS, improved QoL, and an extended treatment-free interval. This underscores the significance of durable response as a valuable endpoint in immunotherapy clinical trials.

In terms of monitoring the virally induced immune response in vivo, more challenges remain. To follow the viral spread, currently clinical samples collected through random biopsies under image guidance can later be analysed for the presence of the virus and immune cell infiltrates in a laboratory. However, this provides a limited representation of the viral replication in the tumour as the sample is taken from one location and the virus can replicate and spread unevenly to distant sites. Frequent sampling or sampling from distinct locations of the tumours was suggested, though these methods would not be ethically appropriate due to the high patient burden resulting from the invasiveness of the methodology [92]. With the advent of single-cell technologies, multiplexed-immunofluorescent images or in situ spatial transcriptomics the sampling burden and the need for the repetitive sampling of the tumour, blood and CSF could be reduced as more detailed information about the viral dynamics, distribution and spatial interactions could be identified with the use of those techniques as they require smaller and fewer samples compared to the information they provide [106,107,108].

Furthermore, monitoring the virally induced immune response through systemic measurement of increases in specific immune cell types does not necessarily reflect the immune status in the TME. Solely quantifying the infiltrating immune cells in the TIME as a marker of effective anti-tumour response is inadequate, due to the diverse pro-tumour and anti-tumour functions of different immune cell populations [109]. As an example, the distribution of virus-specific versus tumour-specific infiltrating T cells is not well characterised, nor investigated during the different stages of treatment in the patients, despite the fact that this ratio could substantially alter the outcome in tumour response [77,109].

Hence, to comprehensively understand the antitumor impact of oncolytic virotherapy without the need for invasive tissue sampling, non-invasive imaging should encompass viral monitoring, assessment of immune checkpoint expression, and the tracing of immune cells infiltrating tumours. To combat these issues, OVs encoding transgenes for reporter genes to enable real-time tracking of viral replication, radiolabelled antibodies to a variety of checkpoint proteins and radiolabelled CAR-T cells are all examples of techniques currently under development to improve our understanding of how virotherapy affects the TIME in vivo [110]. Until, their successful implementation, collection of cerebrospinal fluid (CSF) and tumour biopsies pre- and post-treatment along with several blood samplings during the treatment is highly recommended for the investigation of the OV progression and its effects on the immune modulation as data of this type are severely missing or lacking in most OV clinical trials.

### 6.3. The Interplay between Administration, Neutralising Antibodies and Anti-Viral Response

Multiple factors surrounding the context of administration can additionally affect whether OV treatment is successful in tumour eradication or not. IV administration is always preferred over IT due to the reduced invasiveness and strain it induces on the patient. Upon administration of OVs, the conflict between the host immune system and the virus begins. During OV-infected tumour cell death, along with TAAs, viral antigens are released instigating a parallel anti-viral response. Thus, rapid clearance of the therapeutic OVs by the innate immune system can minimise viral replication and continuous infection in tumour cells [111]. Antiviral cellular immunity and neutralising antibodies constitute the natural immune response against a perceived pathogen.

To partly mitigate this conflict, direct intratumoral administration can be used to maximise OV tumour distribution. However, in cases of metastatic disease or tumours where intratumoral administration is completely or almost completely inaccessible, another strategy is required. To limit the host’s response against the OV and enable repeated administrations, an immunosuppressive agent such as cyclophosphamide can be administered for a short term to suppress innate immune responses and reduce the number of neutralising antibodies [112]. However, neutralising antibodies play a significant role in OV safety as in exceptional cases the OVs can be distributed to organs or body parts away from the tumour area and cause toxicity due to dissemination of the OV in the healthy cells [112].

Interestingly, seroconversion of neutralising antibodies against the OV is also an important factor capable of impacting the efficacy of OV therapies. This often occurs if the patient has had previous exposure to the virus type of the OV but can occur during OV treatment as well. One small benefit is that pre-existing neutralising antibodies (for some virus serotypes) do occur a lot more in adult patients as opposed to paediatric patients [113]. When comparing the MOS of OV-treated DMG patients with low versus high neutralising antibody titres, the patients with low antibody titres had a much more favourable prognosis [15]. A recent clinical trial using an HSV-1 named CAN-3110 demonstrated the opposite results. During that trial patients with recurring GBM were treated with a single dose of CAN-3110 and, interestingly, existing HSV-1 seropositivity prior to treatment was correlated to higher CD4 and CD8 T cell infiltration intratumorally and a higher clinical benefit when compared to seronegative patients prior to treatment [44]. In this trial, it was highlighted that the responders with seropositivity had no positive staining for HSV-1 intratumorally, while the non-responders with negative HSV-1 seropositivity had positive staining for HSV-1 [44]. This indicated that the tumour in the responders was cleared due to an active anti-viral response, thus the lack of HSV-1 staining, which led to an anti-tumour response [44], whereas the non-responders had weak to no anti-viral response that allowed the OV persistence [44].

Potential ways to circumvent the mechanisms of OV clearance are to (a) introduce multiple-timepoint OV injections to increase the chance of the OVs inducing their effect and switch their seropositivity status, or (b) alter the virus packaging to conceal the OV from the host immune system until tumour cell infection. One of such shielding mechanisms is to package the virus inside carrier cells such as neural stem cells (NSCs) or mesenchymal stem cells (MSCs). These cells have inherent tumour tropism, can cross the BBB, and distribute easily throughout the tumour, whilst enabling the OVs to evade immune cells and neutralising antibodies [42,114]. A clinical trial for multiple adult tumours, including two paediatric MB patients, employing the oncolytic virus Icovir-5 loaded into autologous MSCs (NCT01844661) and a paediatric clinical trial loading Icovir-5 into heterologous MSCs (NCT04758533) already exist; however, the unknown serological status of the patients and a small number of paediatric patients do not provide sufficient evidence for the implication of an anti-viral response in the responders/non-responders [115]. Other strategies to camouflage the OVs are using polymer coatings such as polyethylene glycol (PEG) to limit antibody binding or modifying the virus by switching viral envelopes or capsids of multiple virus serotypes [116]. All these measures would serve as a temporal form of protection for the OVs to be able to initiate their manipulation of the TIME without the need to improve their virulence.

While the magnitude of the innate immune response triggered by OVs is crucial for initiating an adaptive immune response to the virus and potentially in combating tumours, it can also be detrimental to pre-existing effector and memory T cells, including adoptively transferred CAR-T cells. This occurs because these cells are dependent on a specific balance of immunomodulatory mediators for their proliferation and activation in the TIME [77]. Although the combinational therapy of OVs with other modalities of immunotherapies has generated promising preliminary results, the timing and dosage of OV therapy need to be carefully optimised. For instance, as opposed to co-administration of the two therapies, OVs could be used as an ‘immune-priming’ treatment followed up by the second immunotherapeutic modality after the first surge of the inflammatory response has subsided. This model of administration could be used with the multiple potentially synergistic combinations of OV therapy with immunotherapies such as ICIs, adoptive cellular therapy or cancer vaccines that we previously described [13,77].

## 7. Conclusions

Considering the small number of patients and clinical trials tested, the data currently available on OV therapy have demonstrated improved MOS with limited implications; however, an investigation is still required to monitor any long-term effects of this therapy in survivors, a factor that should be implemented in future clinical trials. Based on the preclinical and clinical observations, the highlight of OV action has switched from oncolysis to immunomodulation and thus, the development of OVs that effectively but precisely modulate the TIME to facilitate optimal tumour killing is well underway. Combinations of various immune therapies with OVs are conceived frequently, producing favourable effects, but some roadblocks still exist that hamper their successful implementation. Improvements should be made in terms of dose/effect response, administration routes, number of injections and timing of OV therapy, especially when used alongside other immunotherapy modalities. To facilitate this, pre-clinical models should be developed that accurately reflect the TIME in paediatric patients. Last, the markers of progression should be developed to track effective treatment, to reflect OV persistence/clearance, OV distribution and immunomodulatory effects in the least invasive manner (Figure 5). The first steps towards improvement have been taken and we should soon begin to see the full potential of OV therapy in the treatment of PBTs.

## Figures and Tables

**Figure 1 ijms-25-05007-f001:**
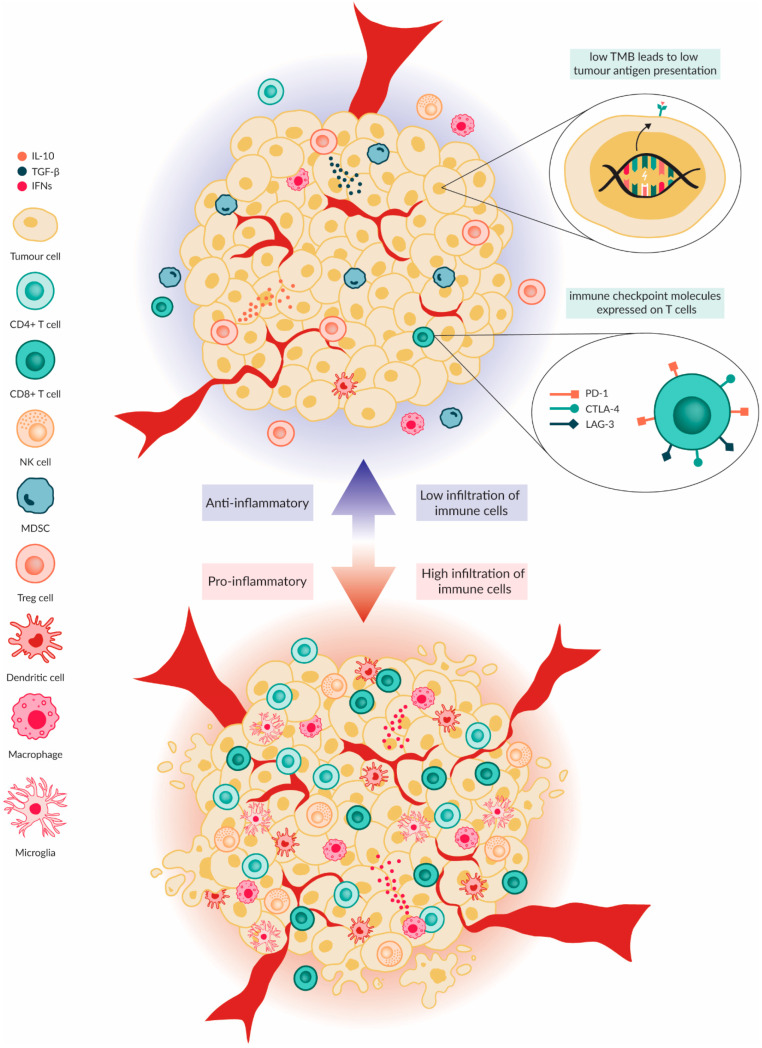
Immune cold versus immune hot tumour microenvironment. The TIME of PBTs is variable depending on the type of tumour; however, it is generally considered to be immunologically cold. This is characterised by low infiltration of CD4+ and CD8+ T cells combined with high infiltration of regulatory T cells and MDSCs. The Tregs and MDSCs secrete immunosuppressive mediators such as IL-10 and TGF-β. The CD4+ and CD8+ T cells that are present in the TIME express high levels of immune checkpoint molecules such as PD-1. Moreover, the TMB in PBTs is low. This results in less tumour-associated and specific antigens being presented on the tumour cell surface for immune cell recognition. Macrophages and microglia present in the TIME are predominantly M2 polarised and, therefore, are tumour supporting. Conversely, in a hot TIME the infiltration of effector T cells, DCs, macrophages and microglia is high. Moreover, a high TMB leads to a higher neoantigen load, resulting in more tumour recognition and immune cell activation.

**Figure 2 ijms-25-05007-f002:**
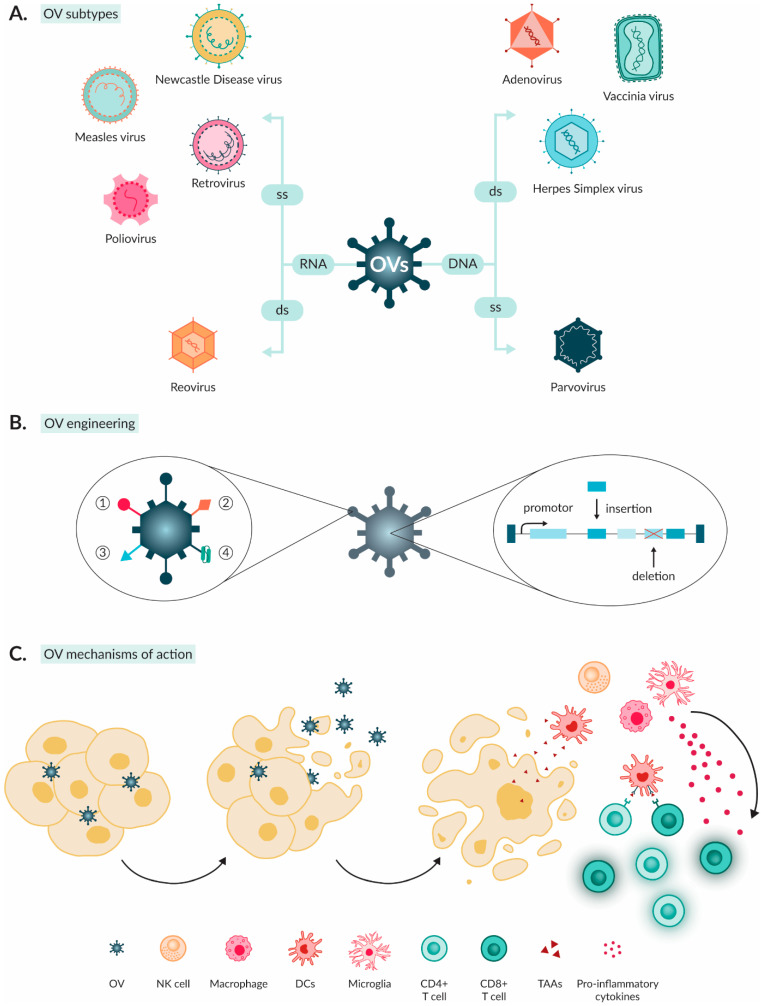
Oncolytic viruses (OVs) for brain tumour treatment. (**A**) There is a broad array of OVs that have been studied as anticancer agents in brain tumours. Different subtypes include RNA and DNA viruses with either single-stranded (ss) or double-stranded (ds) nucleic acids and the presence or absence of a viral envelope. (**B**) Tumour tropism and entry targeting of OVs can be enhanced in multiple ways: (1) ‘serotype switching’, (2) modifying targeting peptides of fibre knob domain, (3) insertion of viral envelope glycoproteins from alternate viral families and (4) insertion of genes encoding for a single-chain antibody specific to a known tumour-surface antigen. Moreover, post-entry tumour specificity can be enhanced through insertion of tumour-specific promotors, which only allow for expression of viral genes in tumour cells or through mutation or deletion of specific genes to deprive OVs of the ability to replicate in normal cells and reduce their virulence. (**C**) OVs’ anti-tumour effect is accomplished through two mechanisms. First, OV infection into tumour cells, resulting in replication and cell lysis to further spread newly synthesised viral particles. Second, the lysed tumour cells release TAAs and damage signals which, along with the virus antigens, induce local APCs to activate and, in turn, activate effector T cells such as CD4+ and CD8+ T cells.

**Figure 3 ijms-25-05007-f003:**
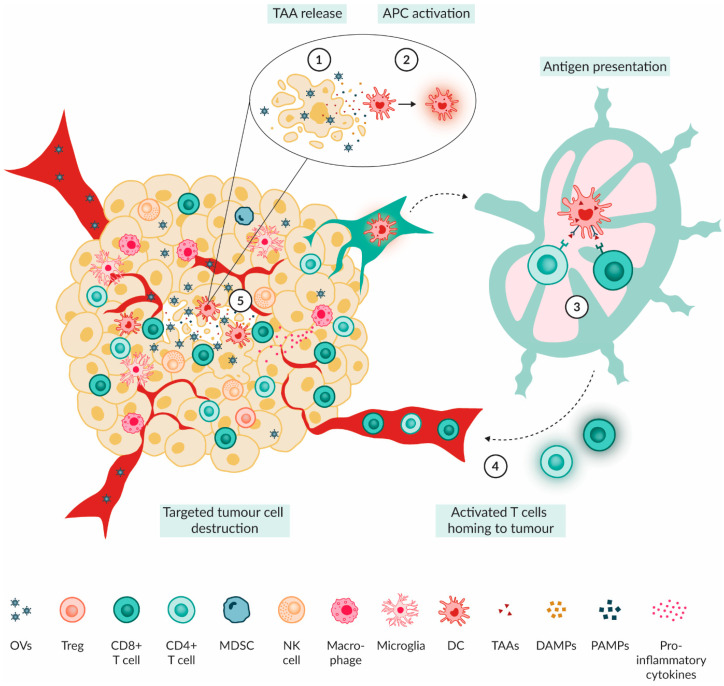
The OV-mediated modulation of the tumour immune microenvironment. (1) Upon OV-induced tumour cell lysis, TAAs, cytokines, DAMPs and PAMPs are released into the TME. (2) Innate immune cells such as NK cells, macrophages and DCs get activated by these released particles. NK cells will directly lyse tumour cells. (3) Upon activation, DCs will act as APCs and migrate to the lymph nodes. There, they can present the TAAs to T cells, leading to the activation of CD4+ and CD8+ T cells. (4) The activated effector T cells travel to the tumour site with the help of secreted cytokines and chemokines by macrophages and DCs. (5) Upon arrival at the tumour site, CD8+ T cells selectively target and kill tumour cells that present the TAAs.

**Figure 4 ijms-25-05007-f004:**
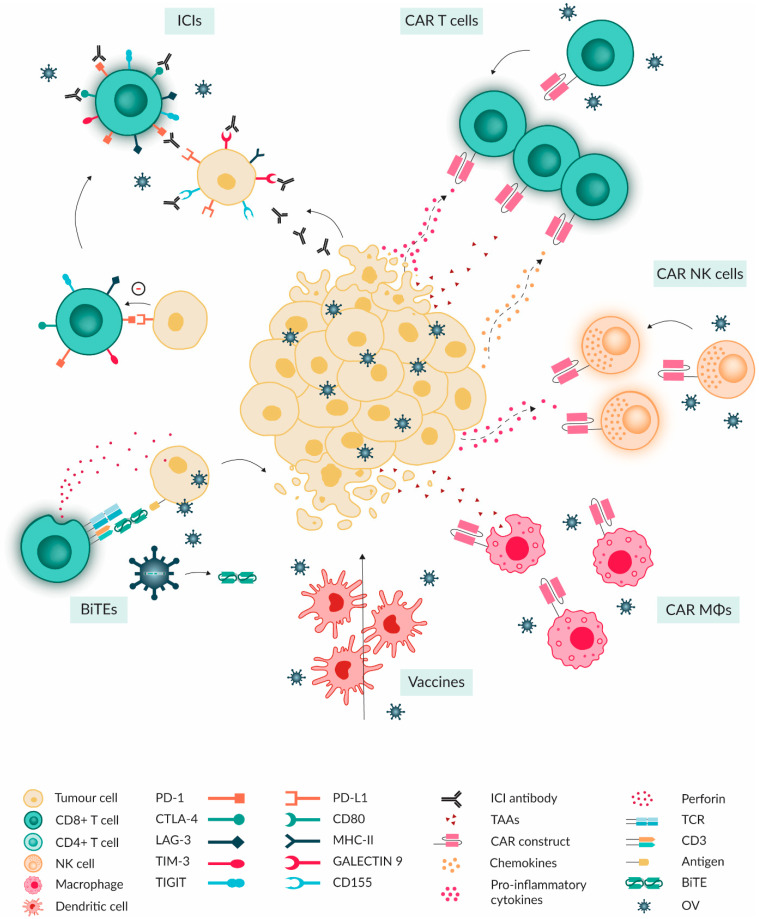
Strategies for combination therapy of OVs with immunotherapies. The combined tumour targeting with both OVs and different immunotherapy modalities could lead to more efficient tumour killing. OV therapy has been shown to lead to upregulation of immune checkpoint molecules on both tumour and immune cells, thereby priming the tumour for effective ICI therapy. Furthermore, OV infection incites the release of cytokines and chemokines. This can help recruit CAR T and NK cells to the tumour site and stimulate proliferation. Modified OVs can express target ligands for CAR macrophages or ligands to block tumour escape from phagocytosis. Moreover, the OV-induced influx of cytotoxic T cells and T helper cells can improve the efficacy of cancer vaccines. Last, BiTEs can be produced at the tumour site by encoding their transgenes into OVs, providing improved delivery. CD80: cluster of differentiation 80, MHC-II: major histocompatibility complex II, TIM-3: T cell immunoglobulin and mucin domain-containing protein 3, TIGIT: T-cell immunoreceptor with Ig and ITIM domains, CD155: cluster of differentiation 155, CAR: chimera antigen receptor, TCR: T cell receptor, CD3: cluster of differentiation 3.

**Figure 5 ijms-25-05007-f005:**
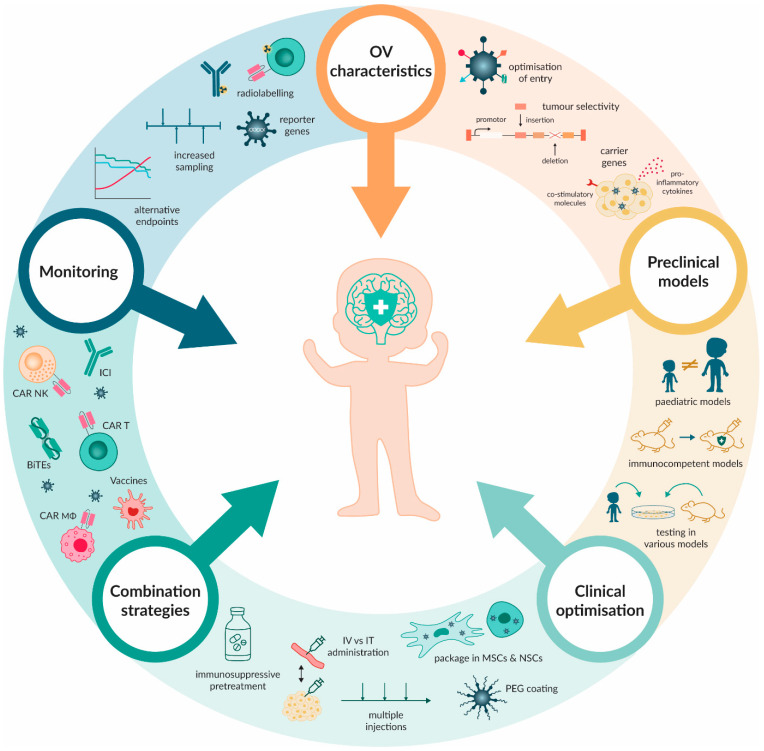
Overview of future suggestions to improve OV therapeutic efficacy, safety and monitoring. OV characteristics affecting entry and safety can be optimised through serotype switching, altering fibre knobs, capsid proteins or targeting surface antigens. Moreover, OVs can be genetically modified to only allow replication in tumour cells. To enhance OV-induced tumour cell eradication, OVs can carry transgenes inducing the expression of costimulatory molecules or secretion of pro-inflammatory cytokines by infected tumour cells. Current preclinical models of PBTs can be optimised by developing specific paediatric models (as opposed to adult), immunocompetent models, and by using of a variety of models to improve the translation of preclinical data to the clinic. During clinical use of OVs, modifications can be made to improve the OV efficacy. The perseverance of OVs in the body can be enhanced by using polymer coatings or packaging the OVs in carrier cells that can cross the BBB. Moreover, the manner and timing of administration can be altered as well as the consideration of pretreatment with an immunosuppressive agent to minimise the innate immune response targeting the OVs. OV therapy can additionally be combined with other immunotherapeutic modalities, potentially leading to synergetic therapeutic effects. The monitoring of OV therapy can be improved by focusing on alternative endpoints to avoid unnecessary early termination of treatment, by increasing sampling intervals, and by improving methods of continuous non-invasive monitoring. MSC: mesenchymal stem cell, NSC: neural stem cell, IV: intravenous, IT: intratumoral.

**Table 1 ijms-25-05007-t001:** Oncolytic viruses investigated in clinical trials for the treatment of paediatric and adult brain cancers. IT: intratumoural administration, IV: intravenous administration, IA: intra-arterial administration, ICT: intracavitary administration, LB: lumbar puncture, MOS: median overall survival, OSR: Overall survival rate. Paediatric trials are indicated in orange, adult trials are indicated in green.

Virus Type	Name	Modifications	Clinical Trial (Administrative Route)	Paediatric or Adult	Patients (N)	Trial Status	Survival	Ref.
	DNA viruses							
Adenovirus (AdV)	DNX-2401	Insertion of RGD-4C peptide in the fiber knob. 24 bp deletion in E1A viral gene responsible for Rb-binding.	#NCT03178032 (IT)	Paediatric	12	Completed	MOS = 17.8 months/OSR = 50% at 18 moths	[15]
			#NCT02197169 (IT)	Adult	27	Completed	OSR = 33% at 12 months, 22% at 18 months	[39]
			#NCT00805376 (IT)	Adult	37	Completed	MOS = 12 months	[40]
			#NCT01956734 (IT)	Adult	31	Completed	Not released yet	
			#NCT01582516 (IT)	Adult	20	Completed	MOS = 4.3 months	[41]
			#NCT02798406 (IT)	Adult	49	Completed	MOS = 12.5 months/OSR = 52.7% at 12 months	[19]
			#NCT03896568 (IA)	Adult	36	Recruiting	-	
	Ad-TD-nsIL12	Deletion in E1A, E1B and E3gp-19k genes. Expresses a non-secretory form of IL-12 under E3gp-19k promoter.	#NCT05717712 (IT)	Paediatric	18	Recruiting	-	
			#NCT05717699 (IT)	Paediatric	18	Recruiting	-	
	DNX-2440	Insertion of RGD-4C peptide in the fibre knob. 24 bp deletion in E1A viral gene responsible for Rb-binding. Expresses the co-stimulatory OX40 ligand, replacing E3 region.	#NCT03714334 (IT)	Adult	16	Terminated	Not yet released	
	CRAd-S-pk7	Deletion of native E1 promoter. E1A expression under control of an inserted human surviving promoter and additionally encodes pk7 polylysine.	#NCT05139056 (ICT)	Adult	36	Recruiting	MOS = 15.7	
			#NCT06169280 (IT)	Adult	-	Not recruiting yet	-	
			#NCT03072134 (IT)	Adult	12	Completed	MOS = 18.4 months	[42]
	ONYX-015	Deletion of E1B-55kD gene.	#NCT00006106 (IA)	Adult	24	Withdrawn	MOS = 6.2 months	[43]
	ICOVIR-5	Insertion of RGD-4C peptide in the fibre knob. 24 bp deletion in E1A viral gene responsible for Rb-binding. In addition, the E1A promoter is replaced by E2F1-responsive elements.	#NCT04758533 (IV)	Paediatric	12	Recruiting	-	
Herpes simplex virus (HSV)	C134	Deletion of both copies of the principal virulence gene γ_1_34.5. Additionally has IRS1 gene under control by human cytomegalovirus immediate early promoter.	#NCT03657576 (IT)	Adult	24	Recruiting	-	
	G207	Contains deletion of the diploid γ_1_34.5 neurovirulence gene and has viral ribonucleotide reductase (UL39) disabled by insertion of Escherichia coli lacZ.	#NCT02457845 (IT)	Paediatric	12	Completed	MOS = 12.2 months	[16]
			#NCT03911388 (IT)	Paediatric	24	Active, not recruiting	-	
			#NCT04482933 (IT)	Paediatric	40	Not yet recruiting	-	
	M032	Contains deletion of the diploid γ_1_34.5 neurovirulence gene. Expresses the gene for IL-12.	#NCT02062827 (IT)	Adult	29	Active, not recruiting	-	[20]
	CAN-3110	Deletion of one copy of the principal virulence genes γ_1_34.5 and UL39. ICP-34.5 under control of human Nestin promoter.	#NCT03152318 (IT)	Adult	62	Recruiting	MOS = 11.6 months	[44]
	G47Delta	Deletion of both copies of the principal virulence gene γ_1_34.5. Inactivation of the ICP6 gene. Deletion of the a47gene and US11 promoter.	#UMIN000015995 (IT)	Adult	30	Completed	MOS = 20.2	[45]
	HSV1716	759bp deletion in both copies of the principal virulence gene γ_1_34.5.	#NCT02031965 (IT)	Paediatric	2	Terminated	Not released yet	
Vaccinia Virus (VV)	TG6002	Deletion of viral thymidine kinase (TK) gene. Deletion of the viral ribonucleotide reductase (RR) gene. Addition of chimeric yeast FCU1 suicide gene in TK locus.	#NCT03294486 (IV)	Adult	78	Unknown	Not yet released	
Parvovirus (PVV)	ParvOryx	Unmodified.	#NCT01301430 (IT/IV)	Adult	18	Completed	MOS = 15 months	[46]
	RNA viruses							
Reovirus (RV)	Pelareorep	Unmodified.	#NCT02444546 (IV)	Paediatric	6	Completed	MOS = 11.7 months	[17]
			#NCT01166542 (IV)	Adult	167	Completed	Not released yet	
Newcastle disease virus (NDV)	NDV-HJU	Unmodified.	#NCT01174537 (IV)	Paediatric and Adult	30	Withdrawn	Not released yet	[47]
Measles virus (MV)	MV-NIS	Expresses human thyroidal NIS.	#NCT02962167 (IT/LB)	Paediatric	34	Completed	Not yet released	
	MV-CEA	Secretes the extracellular domain of human CEA.	#NCT00390299 (IT)	Adult	23	Completed	MOS = 11.6/OSR = 45.5% at 12 months	[45]
Poliovirus (PV)	PVSRIPO	Live-attenuated poliovirus vaccine carrying a heterologous internal ribosomal entry site (IRES) of human rhinovirus type 2 (HRV2).	#NCT03043391 (IT)	Paediatric	8	Completed	MOS = 4.1 months	[18]
			#NCT02986178 (IT)	Adult	122	Active, not recruiting	MOS = 12.5 months/OSR = 21% at 24 and 36 months	[48]

**Table 3 ijms-25-05007-t003:** Overview of the reported OV-induced immune cell modulations in the paediatric tumour microenvironment in a clinical setting. This table only reported OVs with innate immunomodulatory effects.

Immune Cell Type	Modulation	OV Type (Name)	Tumour Type	Ref.
CD4+ T cells	Increased influx in TME	AdV (DNX-2401)	DMG	[15]
	Durable increased influx in TME and adjacent + distant sites	HSV (HSV-1 G207)	HGG	[16]
CD8+ T cells	Increased influx in TME	AdV (DNX-2401), PV (PVSRIPO)	DMG, HGG	[15,18]
	Durable increased influx in TME and adjacent + distant sites	HSV (HSV-1 G207)	HGG	[16]
Tregs	Absent after OV treatment	AdV (DNX-2401)	DMG	[15]
B cells	Increased levels in TME	HSV (HSV-1 G207)	HGG	[16]
Plasma cells	Increased levels in TME	HSV (HSV-1 G207)	HGG	[16]
DCs	Increased levels of monocytes	RV (Pelareorep)	GBM, DMG, MB	[17]
Macrophages	Up-regulation of immune response terms	AdV (DNX-2401)	DMG	[15]
	Increased levels of monocytes	RV (Pelareorep)	GBM, DMG, MB	[17]

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
