# Peer review of "OV Modulators of the Paediatric Brain TIME: Current Status, Combination Strategies, Limitations and Future Directions"

_ijms, 2024, doi:10.3390/ijms25095007_

Round 1
Reviewer 1 Report
Comments and Suggestions for Authors
This manuscript titled "OV modulators of the paediatric brain TIME: current status, combination strategies, limitations and future directions" presents a comprehensive exploration of the current state of research and application of Oncolytic Viruses (OVs) in treating paediatric brain tumours, focusing on their ability to modify the tumour immune microenvironment (TIME), combination therapies, and the challenges that need to be addressed for future progress. This article has certain novelty, but a significant amount of work is needed before consideration for acceptance. Thus, major revision is recommended. Main issues have been listed as followed:
1.Authors must comprehensively discuss and thoroughly compare the OVs with current and standard therapeutic approaches for treating PBT, and list their advantages and disadvantages based on existed evidence, for presenting the potential clinical significance of OV, and the meaning of studying such field.
2. If possible, an update on the status of ongoing clinical trials mentioned in the manuscript, including any preliminary results or findings since the manuscript was written, is highly recommended.
3. This manuscript should incorporate patient-centered outcomes, including quality of life and long-term survival, would provide a more holistic view of the impact of these therapies.
4. How are the clinical outcomes of trials selected in this manuscript. Without actual clinical data, it is extremely difficult to evaluate the advantages of the therapy. This is an unavoidable issue for the article, but it must be properly addressed.
5.The manuscript is dense with technical details and could benefit from simplification and clarification in some sections to make it more accessible to readers not intimately familiar with the field. For instance, explanations of certain mechanisms could be simplified, and the significance of findings could be more clearly highlighted.
6. Data of “4.1 Preclinical evidence” should be listed in a table which includes the information of outcomes, models(cell, animal), year, etc.
7.The discussion on translational limitations and model systems is crucial. This manuscript should include more specific recommendations or emerging technologies that could address these limitations. This might encompass advances in imaging techniques, biomarker discovery, or novel model systems that better recapitulate the paediatric brain tumour microenvironment.
8. Introduction should be more comprehensive explaining each aspect of article.
9. While the manuscript discusses combination therapies, there's room to delve deeper into the reasoning, evidence, and theories guiding the choice of specific combination strategies over others. This could involve a more thorough examination of the reasons behind selecting particular immunotherapies to pair with OVs, as well as the expected or observed synergistic effects.
Author Response
Responses to reviewer 1
Dear reviewer, thank you very much for taking the time to review this manuscript. The authors have carefully reviewed each comment and have addressed them comprehensively. Please find the detailed responses below and the corresponding revisions highlighted in the re-submitted files
1.Authors must comprehensively discuss and thoroughly compare the OVs with current and standard therapeutic approaches for treating PBT, and list their advantages and disadvantages based on existed evidence, for presenting the potential clinical significance of OV, and the meaning of studying such field.
Response: The current and standard therapeutic approaches for PBTs greatly differs by the type, location and age of diagnosis. A thorough discussion of the advantages and disadvantages between those therapies would overextend the introduction. Therefore, we focused on, the most prevalent treatments like surgical resection, chemotherapy and radiotherapy and their overall effect on overall survival and quality of life (Lines 35-47). Moreover, the advantages of OVs are actively discussed throughout the manuscript as great adjuvants for immunotherapies due to their multimodal action and potential clinical significance, as demonstrated in the limited number of trials currently completed in PBTs. However as these findings are scarce, and lack information on the long-term effects in survivors due to the novelty of the therapy, we decided to focus on their combinational potential.
2.If possible, an update on the status of ongoing clinical trials mentioned in the manuscript, including any preliminary results or findings since the manuscript was written, is highly recommended.
Response: Dear reviewer thank you for highlighting the need to include these data. Table 1 has been updated and expanded, mentioning the number of patients included in the trial, the status of the trial, overall survival rate and median overall survival when applicable. The variables included highlight the limited size of the pediatric trials, their ongoing status and the improved survival that was demonstrated in completed trials.
3.This manuscript should incorporate patient-centered outcomes, including quality of life and long-term survival, would provide a more holistic view of the impact of these therapies.
Response: Thanks to your inquiry, outcomes of survival have been included when possible and relevant throughout the manuscript. A small paragraph summarizing the lack of OV treatment related severe adverse effects, neurological deterioration or decline on the quality of life of the patients was also added. Lines (366-369) However, it is important to stretch that very little information is currently available on long term survival and QOL due to the recent completion of the trials, a very important aspect that should be incorporated in future trials.
4.How are the clinical outcomes of trials selected in this manuscript. Without actual clinical data, it is extremely difficult to evaluate the advantages of the therapy. This is an unavoidable issue for the article, but it must be properly addressed.
Response: All clinical outcomes stated are up to date with the submission of the manuscript. As our main focus was to highlight the use of OV as an adjuvant therapy with other immunotherapies, we mainly focused on describing the interactions of OVs with the TIME, which in all clinical trials seemed to be linked to the improved survival of the patients. Whenever available the limited clinical evidence from paediatric trials (Table 1 and Table 3) and adult trials were described merely to demonstrate the benefit of combinations with existing immunotherapies as no relevant paediatric trials exist yet.
5.The manuscript is dense with technical details and could benefit from simplification and clarification in some sections to make it more accessible to readers not intimately familiar with the field. For instance, explanations of certain mechanisms could be simplified, and the significance of findings could be more clearly highlighted.
Response: Thank you for highlighting the need for simplification of the text. In order to accommodate your comment we simplified the technical details across the manuscript, but still described the interactions of OVs with the TIME in detail, as they are the main focus of the manuscript. However, some newly introduced mechanisms had to be introduced and explained more extensively. In addition Figure 1-3 helps to alleviate the complex information covered in the review with visual illustrations and the figure legends.
6.Data of “4.1 Preclinical evidence” should be listed in a table which includes the information of outcomes, models(cell, animal), year, etc.
Response: Table 2 previously summarized the interactions of OVs with the different immune cell types of the TIME. However based on your comment we decided to split the table into two individual tables (Table 2 and 3). The new Table 2 now focuses on the preclinical evidence including survival outcomes, the tumor cell lines and types, the animal models used and the TIME interactions, while table 3 focuses mainly on the clinical evidence of the OV interactions with TIME.
7.The discussion on translational limitations and model systems is crucial. This manuscript should include more specific recommendations or emerging technologies that could address these limitations. This might encompass advances in imaging techniques, biomarker discovery, or novel model systems that better recapitulate the paediatric brain tumour microenvironment.
Response: The discussion on translational limitations and model systems was expanded by adding some recommendations on newer technologies, including humanized mouse models and patient-derived in vitro models. Lines (647-710)
8.Introduction should be more comprehensive explaining each aspect of article.
Response: The introduction was slightly expanded to include the different aspects of the manuscript. We decided to keep some introductory aspects of the manuscript short to avoid repetition with the main parts of the manuscript that describe the aspects mentioned in depth. However, we included some additional introductory information on the general efficacy of conventional therapies and their adverse effects on neurological developments as they are not extensively discussed in the main parts of the manuscript. Lines (35-96)
9.While the manuscript discusses combination therapies, there's room to delve deeper into the reasoning, evidence, and theories guiding the choice of specific combination strategies over others. This could involve a more thorough examination of the reasons behind selecting particular immunotherapies to pair with OVs, as well as the expected or observed synergistic effects.
Response: Thank you for pointing out your concern. All the information described throughout the manuscript until chapter 5 is used as an introduction for the reasoning and evidence for using OVs in combination with existing immune therapies. The manuscript covers as many combination strategies as possible that are currently being investigated, without biasing the reader towards specific strategies. This is taking into consideration the future of therapies around personalized treatment where not every therapy can successful treat all types of tumours or all patients, thus having access to a plethora of therapies might lead to better responses based on each case.
Reviewer 2 Report
Comments and Suggestions for Authors
The review is clearly structured, well-written, with excellent content, illustrations and tables. While there are other reviews of this subject, the structure and content of this manuscript make it a useful addition to the literature.
Comments on the Quality of English LanguageThe English is well-written. There are rare grammatical/syntactical errors, but insignificant, and the flow is good.
Author Response
Response to reviewer 2
Dear Reviewer thank you for taking the time to review this manuscript. We identified and fixed any grammatical errors we could detect. You can find all revisions and changes requested by the reviewers highlighted in the submitted file.
Round 2
Reviewer 1 Report
Comments and Suggestions for Authors
Thank the authors for their revision. The manuscript now provides a comparatively detailed and informative review of the role of OV in pediatric brain tumor immunotherapy. With improvements in data presentation, detailed discussion of translational challenges, and expanded future directions, this review article can help researchers and clinicians to better understand and potentially overcome the hurdles in treating pediatric brain tumors with OVs.